# Over twenty years of publications in Ecology: Over-contribution of women reveals a new dimension of gender bias

Gabriela Fontanarrosa[1]☯, Lucía Zarbá[2]☯, Valeria Aschero[3], Daniel Andrés Dos Santos[1,4], María Gabriela Nuñez Montellano[5], Maia C. Plaza Behr[5], Natalia Schroeder[6,7], Silvia Beatriz Lomáscolo[5], María Elisa Fanjul[4,8], A. Carolina Monmany Garzia[5], Marisa Alvarez[9,10], Agustina Novillo[1], María José Lorenzo Pisarello[11], Romina Elisa D'Almeida[12], Mariana Valoy[8], Andrés Felipe Ramírez-Mejía[5], Daniela Rodríguez[6,7], Celina Reynaga[1], María Leonor Sandoval Salinas[13,14], Verónica Chillo[15], María Piquer-Rodríguez[16]*

1 Instituto de Biodiversidad Neotropical (IBN), Consejo Nacional de Investigaciones Científicas y Técnicas (CONICET), Facultad de Ciencias Naturales e Instituto Miguel Lillo, Universidad Nacional de Tucumán (UNT), Yerba Buena, Tucumán, Argentina, 2 Instituto de Investigaciones Territoriales y Tecnológicas para la Producción del Hábitat UNT-CONICET, Tucumán, Argentina, 3 Instituto Argentino de Nivología, Glaciología y Ciencias Ambientales (IANIGLA), CONICET, Universidad Nacional de Cuyo (UNCuyo), Mendoza, Argentina, 4 Instituto Vertebrados, Zoología, Fundación Miguel Lillo, Facultad de Ciencias Naturales e Instituto Miguel Lillo, Universidad Nacional de Tucumán, Yerba Buena, Argentina, 5 Instituto de Ecología Regional (IER), Universidad Nacional de Tucumán (UNT)- Consejo Nacional de Investigaciones Científicas y Técnicas (CONICET), Yerba Buena, Tucumán, Argentina, 6 Instituto Argentino de Investigaciones de las Zonas Áridas (IADIZA), CCT-CONICET, Argentina, 7 Facultad de Ciencias Agrarias, Universidad Nacional de Cuyo, Mendoza, Argentina, 8 Fundación Miguel Lillo, Tucumán, Argentina, 9 Universidad Nacional de Tucumán, Argentina (UNT), Argentina, 10 Universidad Nacional de Santiago del Estero, Argentina (UNSE), Argentina, 11 Centro de Referencia para Lactobacilos CCT NoA Sur. Consejo Nacional de Investigaciones Científicas y Técnicas (CONICET), Argentina, 12 Instituto Superior de Investigaciones Biológicas (INSIBIO), CCT NoA Sur. Consejo Nacional de Investigaciones Científicas y Técnicas (CONICET), Argentina, 13 Instituto de Investigación en Luz, Ambiente y Visión (ILAV), CONICET-UNT, Argentina, 14 Instituto de Investigaciones en Biodiversidad Argentina (PIDBA), Universidad Nacional de Tucumán (UNT), Yerba Buena, Tucumán, Argentina, 15 Instituto de Investigaciones Forestales y Agropecuarias Bariloche (IFAB) IFAB INTA-CONICET, Agencia de Extensión Rural de El Bolsón, Argentina, 16 Institute of Geographical Sciences, Freie Universität Berlin, Berlin, Germany

☯ These authors contributed equally to this work.
* maria.piquer-rodriguez@fu-berlin.de

**Data Availability Statement:** Raw data is available in Fontanarrosa, G.; Zarbá, L.; Aschero,V; Dos Santos, DA; Nuñez Montellano, M.G.; Plaza Behr,

## Abstract

Biographical features like social and economic status, ethnicity, sexuality, care roles, and gender unfairly disadvantage individuals within academia. Authorship patterns should reflect the social dimension behind the publishing process and co-authorship dynamics. To detect potential gender biases in the authorship of papers and examine the extent of women's contribution in terms of the substantial volume of scientific production in Ecology, we surveyed papers from the top-ranked journal *Ecology* from 1999 to 2021. We developed a *Women's Contribution Index* (WCI) to measure gender-based individual contributions. Considering gender, allocation in the author list, and the total number of authors, the WCI calculates the sum of each woman's contribution per paper. We compared the WCI with women's expected contributions in a non-gender-biased scenario. Overall, women account for 30% of authors of *Ecology*, yet their contribution to papers is higher than expected by chance

M.,; Schroeder, N.; Lomáscolo, S.; Fanjul, M.E;
Monmany Garzia, C.; Alvarez, M.; Novillo, A.,;
Lorenzo Pisarello., M.J, D'Almeida, R.E;, Valoy, M.;
Ramírez-Mejía, A.F; Rodríguez, D.; Reynaga, C.;
Sandoval Salinas, M.L.; Chillo, V. & Piquer-
Rodríguez. Ecology Authorships Gender 1999-
2021. Fighare. 2024. 10.6084/m9.figshare.
25953058.

**Funding:** This research received funding from the
Freie Universität Berlin (https://www.fu-berlin.de/)
awarded to MPR; and Agencia (http://www.
agencia.mincyt.gob.ar/): PICT 2019-4546 awarded
to GF and PICTO Género 0022-2022 awarded to
GNM. Sponsors or funders did not play any role in
the study design, data collection and analysis,
decision to publish, or preparation of the
manuscript.

**Competing interests:** The authors have declared
that no competing interests exist.

(i.e., over-contribution). Additionally, by comparing the WCI with an equivalent *Men's Contribution Index*, we found that women consistently have higher contributions compared to men. We also observed a temporal trend of increasing women's authorship and mixed-gender papers. This suggests some progress in addressing gender bias in the field of ecology. However, we emphasize the need for a better understanding of the pattern of over-contribution, which may partially stem from the phenomenon of over-compensation. In this context, women might need to outperform men to be perceived and evaluated as equals. The WCI provides a valuable tool for quantifying individual contributions and understanding gender biases in academic publishing. Moreover, the index could be customized to suit the specific question of interest. It serves to uncover a previously non-quantified type of bias (over-contribution) that, we argue, is the response to the inequitable structure of the scientific system, leading to differences in the roles of individuals within a scientific publishing team.

## Introduction

Meritocracy is a theoretical social system of personal advancement, promotion, and recognition, depending exclusively on a combination of individual attributes: training, talent, and effort, i.e., *merit* [1, 2]. The concept of meritocracy emerged as a contrast to aristocracy, suggesting that a person's position in society should be based on their achieved merits rather than their inherited familial status [1]. The scientific system explicitly aims to be meritocratic, objective, and neutral, applying mechanisms and policies to guarantee this, such as thorough evaluation by peers of demonstrated achievements and capacities [3]. However, the assumptions of meritocracy in terms of equal opportunities and fair competition [2, 4, 5], are not fulfilled in science due to the proven existence of inequalities that interfere with the chances of goal achievement [6–8]. Moreover, the notion of meritocracy disregards the historical and political aspects of individuals' circumstances and therefore may justify inequalities [1].

Inequalities in science negatively affect people based on their identity and biographic features such as social and economic status, ethnicity, sexuality, care tasks, and gender, among others, and the intersectionalities among those attributes [1, 7, 9]. In particular, gender bias results from multiple interactions and feedback loops that occur across various scales, ranging from individual and family levels to workplaces and societal structures [10]. In academia, the existence of gender bias is particularly well-supported by a growing number of studies documenting differential barriers for women across the globe [10–12] affecting well-being perception, productivity (i.e., number of papers published) [13, 14], academic impact (i.e., number of citations), career length [14], research team constitution [15–17], and peer recognition [2, 8, 18–21], among others.

Gender bias in academia can be classified into two main types: i) obstacle bias, and ii) requirement bias. Obstacle bias refer to the barriers that women face in their academic careers, such as double burden (i.e., academic work and domestic unpaid work), less intellectual stimulation, less support, fewer role models, and sexual harassment, among others [10, 17, 21, 22]. Requirement bias refers to the commonly implicit higher expectations placed on women's work, recommendations, and hiring evaluations compared to men's, perpetuated by both men and women. In other words, for equal merits, men are better rewarded than women [2, 8, 19, 20] but see [23]. The biases in requirements and the undervaluation of women in comparison to men, despite having equal merits, are two interrelated aspects of gender bias. The

requirement bias translates into an undervaluation of female researchers' contributions within research teams [3]. For women's contributions to be perceived and evaluated as equivalent to men's, it has been suggested that female researchers must outperform them in terms of the amount and quality of papers [2, 19, 20].

Authorship is central to the recognition and reward system within the historically expanding and evolving network of ideas, papers, and scholars´ contributions [24–26] that shapes scientific knowledge. It directly influences researchers' career prospects [27–30] and plays a crucial role in the scientific community. The order of authors in a co-authorship list typically reflects their degree of contribution in terms of time investment to a paper, with the first author contributing the most, and the contribution decreasing with each subsequent position. The last author may or not reflect an advisory role [8, 28, 30]. Nevertheless, a co-authorship position may also imply other or even arbitrary decisions (but see [31]). Despite that, a bibliometric analysis of the authors' inclusion in a paper and their positions may capture key aspects of the social dimension behind the publishing process and co-authorship dynamics [32, 33]. In this study, we present a gender-based bibliometric analysis of authors' contributions to the papers published in the journal Ecology (henceforth *Ecology*) between 1999 and 2021 as a study case of a high-ranking journal in ecology, a Science, Technology, Engineering, and Mathematics (STEM) discipline, where gender bias is reported [14]. Our analysis investigates potential gender biases in the authorship of papers and examines the extent of women's contribution to the volume of scientific production in the ecology discipline.

## Methods

### Study case: The field of ecology and the journal *Ecology*

Ecology is a field of study within biology that focuses on the relationships between living organisms and their physical environment. Ecological studies also provide information about nature's contributions to people and how we can use the Earth's components in a way that maintains a healthy environment for future generations [34]. Within STEM, men's careers in biology are, on average, 19% longer than women's, resulting in a gender bias of total productivity that exceeds 35% [14]. This gender bias in career length is greater in biology than in applied physics, for example [14]. The ecological literature is dominated by male scientists mainly from North America and Europe [9, 29]. The journal *Ecology* is edited by the Ecological Society of America (ESA) and published by Wiley-Blackwell editorial. It was established in the United States in 1920 and has a high impact factor (4.8 for 2022) within the field. Their papers greatly contribute to shaping the global ecological agenda and conceptual framework. The decision to analyze data from the field of ecology was based on several factors: (1) most of the authors posing the research question belong to the discipline of ecology; (2) ecology serves as a representative STEM discipline where men outnumber women in authorship. In many fields, including ecology, women constitute about 30% of all authors [29, 35]; (3) the common convention in ecology, as in other STEM fields, is to assign the first author position to the individual who contributed the most to the study [30]. Therefore, authorship bias in ecology could be indicative of patterns in other STEM disciplines that exhibit similar trends, particularly biology, chemistry, and mathematics, which have been identified as more gender-balanced than other fields [36]. Thus, the community of researchers publishing in *Ecology* provides a suitable model for investigating hierarchical gender bias in science.

### Data acquisition, data curation, and limitations

We surveyed all *Ecology* papers in the categories "Papers", "Reports", "Reviews" and "Special Issues" from 1999 to 2021 (22 years). For each paper we recorded the list of authors and

classified them as "man" or "woman" using the first name as a proxy of gender by checking available databases (such as Gender Checker, 2020, available at: https://genderchecker.com/), and when necessary, by searching for the authors in Google Scholar or their ResearchGate profiles, among other academic social networks [16, 21]. This strategy is more accurate than automatized classifications [37]. We excluded any paper in which the gender of an author could not be identified. Out of 6125 articles in ecology in the surveyed categories, 993 articles were discarded because they contained at least one author whose sex could not be reliably determined, leaving 5132 articles in the dataset we used for the analyses.

We acknowledge that our approach has limitations. Firstly, it is limited by binarism and cannot fully capture the self-perceived gender of authors. Additionally, there may be a bias because the public databases we used do not fully represent a diverse range of nationalities and cultural backgrounds. Nonetheless, we do not anticipate substantial changes to our main results based on a previous report that highlights the dominance of authors in top-ranked ecology journals from the United States, the United Kingdom, Australia, Germany, and Canada, which account for over 75% of the top-publishing authors [9]. Meanwhile, other regions from the Global South, as well as Russia, Japan, and South Korea, are strikingly underrepresented in top-ranked ecology journals [9].

## Data analysis

**Gender data overview.** We employed a battery of descriptive statistics for a general exploration of data structure, focusing on the distribution of genders in the paper's authorships. We explored multiple dimensions of overall data structure, by year and by paper, including the number of authors, authorship, participation, and temporal trends of those variables by gender. **Authors** account for every person that appeared at least once in our data set. **Authorship** accounts for the number of authoring events disregarding the author's identity, therefore, a particular author could account for more than one authorship event, and the number of total authorships is higher than the number of total authors. **Publication instances** represent the number of events in which the same author has participated in different papers. We have classified **paper types** considering their gender composition and author numbers and calculated their frequencies.

**Women's contribution index.** To estimate the relative contributions made by female authors to a given paper, we designed the *Women's Contribution Index* (WCI). The WCI is constructed based on the Harmonic Allocation of Authorship Credit following Hagen [38] (Eq 1). The harmonic counting allocates credits according to authorship position in the author list and the number of co-authors. Here we consider that author credit is a proxy of the contribution in terms of time investment of the $i^{th}$ author in a particular paper *sensu* [38]. The assumption under this approach is that the total publication credit is shared among all co-authors, the first author gets the most credit, and in general, the $i^{th}$ author receives more credit than the $(i +1)^{th}$ author. The greater the number of authors per paper, the less credit per author. For the sum of every author's contributions in the paper to be 1 (i.e., to be normalized), each reciprocal author position (i.e., 1/author position) is divided by the summation of all reciprocal positions (i.e., for a three-author paper: 1/1 + ½ + 1/3).

The harmonic credit for the $i^{th}$ author, $C_i$ (*i* referring to the position along the authors' list) in a particular paper with *N* co-authors (following [38]), is calculated as follows:

$$Ci = \frac{1}{i} / \left[ 1 + \left(\frac{1}{2}\right) + \left(\frac{1}{3}\right) + \left(\frac{1}{4}\right) + \ldots \left(\frac{1}{N}\right) \right] \tag{1}$$

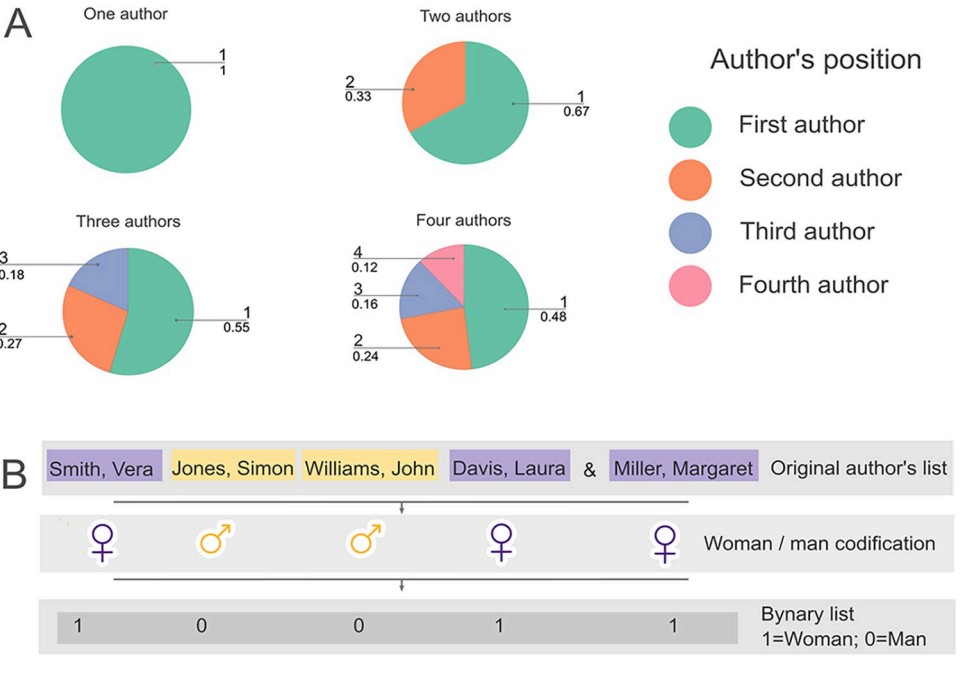

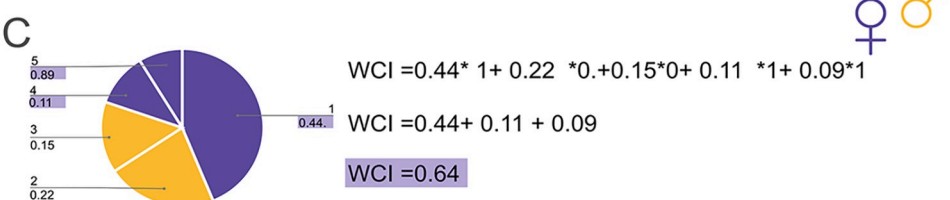

**Fig 1. Graphical representation of the logic behind the Women's Contribution Index (WCI) based on the binary author list in papers following Hagen [38].** A. Encoding of the binary list for a five-authored fictitious paper example. B. Author contribution by positions following the harmonic allocation of authorship credit *sensu* Hagen [38]. C. Women's Contribution Index the calculation for the exemplar paper in B. The index is the sum of the contributions of each woman in the papers. Each woman's contribution is dependent on her position and the whole number of authors. The pie chart depicts a paper of five authors following the exemplar author list in B.

The WCI accounts for the sum of the author´s contribution ($C_i$) of every woman within an author's list (Eq 2). The WCI takes values between 0 (no women contribution) to 1 (complete women contribution). Fig 1 shows an example.

$$WCI = \sum_{i=1}^{N} C_i * G_i \qquad (2)$$

Where:

$C_i$: is the ith author credit *sensu* harmonic allocation [38]

$G_i$: is the gender binary codification of the author of the $i$ position. 1 = woman; 0 = man.

$N$ = is the number of authors in a particular paper

**Women's contribution index in an unbiased scenario.** To test whether women's contributions align with what is expected by chance, we compared the sum of the WCI of the entire dataset (observed total WCI) with its equivalent value in gender-unbiased scenarios (simulated total WCI). We ran 10,000 simulated scenarios where, for each article, we randomly rearranged authors' positions while maintaining the article's gender ratio. We calculated the total

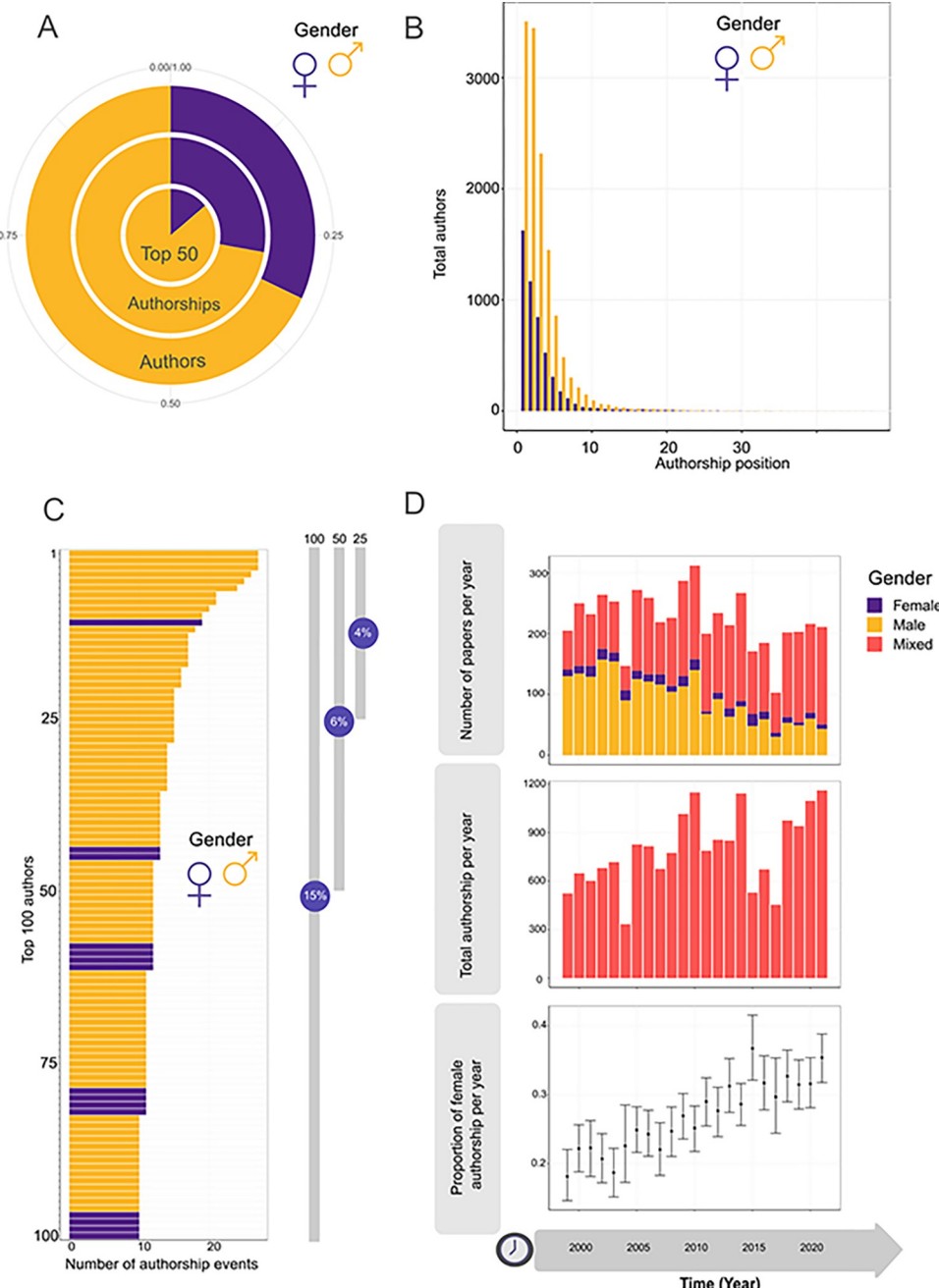

**Fig 2.** Gender Data Overview: A. Pie chart of gender proportion among total authors (outer circle); women proportion among total authorships (middle circle), and women proportion among the 50 authors who published the most (inner circle). B. Absolute frequency histogram of the author's position in the author list discriminated by gender for the entire study period. C. Absolute frequency histogram with the number of papers published by authors during the whole period. The chart shows the top 100 authors who published the most (same author in more than one paper) in decreasing order. The percentage of women among the first, 100, 50, and 25 top authors are shown on the right. D. Upper figure: yearly absolute number of papers published and classified into three categories: exclusively female author lists, exclusively male author lists, and mixed author lists. Middle chart: yearly total authorships in *Ecology*. Lower chart: yearly women's proportion within the authorships.

WCI for each simulation and obtained a distribution of total WCI values, with the mean representing the simulated total WCI. We tested the following statistical hypothesis:

$H_0$ = There is no gender contribution bias within the whole dataset of papers on *Ecology* 1999–2021.

$H_1$ = There is a gender contribution bias within the whole dataset of papers on *Ecology* 1999–2021.

Additionally, we computed the theoretical expected WCI value (expected total WCI) calculated as the sum of the proportions of female authors by article. The expected theoretical expected WCI value depends directly on the women's proportion within the author lists. In a non-biased scenario, the women's contribution index calculated should fit with the expected one.

**Women's contribution index vs. men's contributions index.**   To improve the robustness of our analysis, we compared the WCI to the MCI (Men's Contribution Index) in mixed-gender papers. To avoid dependency on the data we randomly divided the data set into two subsets of equal size. For one subset we calculated the WCI per paper and for the other subset, we calculated the MCI per paper. The MCI followed a procedure identical to that of the WCI, but accounting only for men's contributions. For comparability purposes, we centered the WCI and MCI through the procedure of subtracting their respective expected values (i.e., women's and men's proportions by paper), and we obtained the centered WCI and the centered MCI. Centering is crucial for interpretation when we are interested in group effects [39].

To compare centered WCI vs. centered MCI we performed a quantile-quantile plot (q-q plot). A q-q plot is a plot of the quantiles of the first dataset against the quantiles of the second dataset and is used for diagnosing if two data sets come from populations with a common distribution [40]. Additionally, we performed a Kolmogorov–Smirnov statistical test that quantifies a distance (i.e., dissimilarity) between the empirical distribution functions of two samples (the centered WCI and centered MCI, in our case) [41]. The null distribution of this statistic is calculated under the null hypothesis that the samples are drawn from the same distribution (in the two-sample case).

All the analyses were performed in the *R* environment (R version 3.6.1 [42]) using the *base* and *tidyverse* packages [43]. To preserve the identity of the authors, the database has been encrypted. The encrypted data and [44] executable R code [45] behind all the analysis are available in a permanent repository. Additionally, a printed version of the codes and results are available as Supplementary Information (S1 File). Figures were edited using Inkscape (https://inkscape.org/). To maintain consistency and help readers easily interpret the graphs, we followed the color coding of Grosso et al. [16], purple (RGB:542583ff) for women, yellow (RGB: fcb827ff) for men, and red (RGB:ff2b2aff) for mixed conditions.

## Results

### Gender data overview

A total number of 5,132 papers were analyzed between 1999–2021. Of 11,236 authors in those papers, 3,589 (31.94%) were coded as women. Some authors participated in more than one paper and thus, there were 18,237 authorships, from which 5,074 (27.82%) were coded as women's authorships (Fig 2A). The average publication instances per author was 1.62 (1.41 women, 1.72 men). The average number of authors per paper was 3.55. The four most common paper types consisted of two male-authored papers; 3 mixed-authored papers; 4 mixed-authored papers, and two mixed-authored papers. A list of these and other key numbers are displayed in Table 1.

**Table 1. Key summary values.**

| Key summary variables | Value |
|---|---|
| The average number of authors per paper | 3.55 |
| The average republication value | 1.62 |
| The average republication value of women | 1.41 |
| The average republication value of men | 1.72 |
| Women proportion among the 100 top-publishing authors | 0.15 |
| Women proportion among the 50 top-publishing authors | 0.14 |
| Women proportion among the 25 top-publishing authors | 0.04 |
| Observed total Women's Contribution Index | 1456.08 |
| Expected total Women's Contribution Index (based on female authors per paper) | 1360.49 |
| Mean simulated total Women's Contribution Index | 1360.62 |
| The standard deviation of the simulated total Women's Contribution Index | 7.75 |
| Max simulated total Women's Contribution Index | 1388.23 |
| The most common authorship length in female mono-gender articles | 2 |
| The most common authorship length in male mono-gender articles | 2 |
| Maximum authorship length in female mono-gender articles | 5 |
| Maximum authorship length in male mono-gender articles | 13 |
| Absolute frequency of most common authorship structure: 2 male | 842 |
| Absolute frequency of 2nd most common authorship structure: 3 mixed | 686 |
| Absolute frequency of 3rd most common authorship structure: 4 mixed | 531 |
| Absolute frequency of 2nd most common authorship structure: 2 mixed | 480 |

The table displays the most relevant values extracted from the descriptive statistics and analyses conducted throughout the paper.

Regarding gender position trends in the authors' lists, the most frequent position of female authors was the first one; this frequency monotonically decreased towards backward positions (Fig 2B). Among male authors, both first and second positions shared the higher frequencies in the authors lists, and from the third position backward, frequencies decreased monotonically (Fig 2B).

We observed a strong decrease in the presence of female authors among the authors who published the most (data subsets of 100, 50, and 25 authors were considered). Among the top 100, 50, and 25 authors, 15%, 14%, and 4% were women, respectively (Fig 2C).

Overall, the number of papers authored exclusively by women is very low throughout the studied period. The most notable trend observed is an increase in the number of mixed-gender papers and a decrease in papers exclusively authored by men, with a trend toward reduction (Fig 2D). The maximum authorship length in female mono-gender articles was five, while the maximum authorship length in male mono-gender 13 (Table 1).

Overall, there was an incremental trend in the proportion of women authors, particularly noticeable from 2005 onwards (Fig 2D). Starting around 2012, the ratio of women to total authors appears to stabilize around 0.3 to 0.35. Likewise, the average number of authors per year increased systematically in the surveyed period (Fig 2D).

## *Women's contribution index (WCI)*: **Women's roles in the publishing dynamics**

We measured the WCI for each paper in our dataset and found that the total sum value for all papers was 1,456. Upon conducting the randomized simulations, we obtained a simulated WCI distribution with a mean value of 1,361 and a standard deviation of 7.75. Notably, even

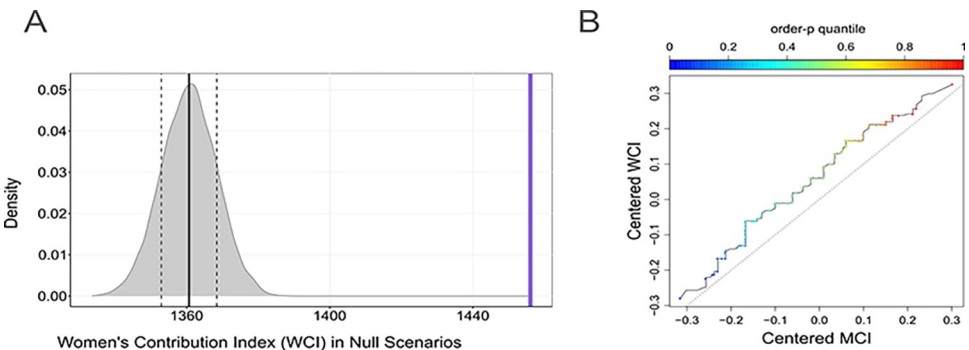

**Fig 3. Women over-contribution to papers.** A. Frequency distribution of the women's contribution index (WCI) in 10,000 random scenarios. The mean value of this distribution is depicted by a thick black vertical line and its standard deviation by dotted vertical lines. The purple vertical line indicates the observed WCI. B. Quantile-quantile plot comparing the cMCI (X-axis) vs the cWCI (Y-axis). The plot displays the pairs of quantiles for probability order quantiles from 0 to 1 (p-order quantiles). p-order quantiles are shown by the colored gradient (blue = 0; red = 1). A 45-degree reference line is also plotted (dotted line) as a reference.

the maximum value in the WCI distribution after running the 10,000 randomized simulations (1,388) did not exceed the observed value of WCI (vertical violet line in Fig 3A, Table 1). Therefore, we could confidently reject the null hypothesis that pointed to no bias in the WCI across the entire volume of *Ecology* papers from 1999 to 2021, for the 10,000 simulated scenarios. None of the simulated scores surpass the observed WCI ($P < 1e-4$). Consequently, the cumulative contributions of women exceeded the expected value by chance.

This observation gained further support when comparing the centered WCI with the centered MCI along the entire rank of p-order quantiles (Fig 3B). The trend demonstrated that the centered WCI consistently exceeded the centered MCI, as indicated by the data points above the reference diagonal line representing women's contributions (Fig 3B). If both data sets came from a population with identical distribution, the points should align along the diagonal reference line. The greater the deviation from this reference line, the stronger the evidence supporting that the two data sets came from different statistical populations. Furthermore, using a Kolmogorov-Smirnov test, we found that the compared distributions were significantly different (D = 0.13971, p-value $< 2.2e-16$). Consequently, we can confidently reject the null hypothesis that the centered WCI and the centered MCI distributions were derived from the same population.

## Discussion

Through the analysis of author lists, we quantified the overall contribution by women in generating the volume of publications of the journal Ecology for over 20 years. Our bibliometric analysis demonstrated that, since 1999, papers in *Ecology* have been mainly men-dominated. Beyond the underrepresentation of women publishing in *Ecology*, our main finding indicated that their contribution, considering the author's position, exceeded what would have been expected by chance. This means that the few women publishing in *Ecology* tend to occupy positions in the author list that require a greater time investment. We referred to this pattern as "over-contribution".

### Measuring gender inequities

To measure gender bias, several indicators have been defined and implemented at different scales [10, 14]. Various studies assessed gender bias in terms of gender disparities in

authorship, number of published papers, citations, or access to funding in almost all disciplines and countries worldwide [11, 13, 32, 46]. However, gender bias is a multidimensional problem rooted in a historical gender imbalance that impacts the success rate of women in academia, therefore, no single indicator is capable of including all its relevant dimensions [10, 20]. For example, women's ratios in the workforce represent the most common tool for diagnosing gender bias, but despite it being a valuable tool it can mask some important dimensions of inequalities such as team members' roles [16, 21, 47]. Grosso et al. [16] used graph theory and found that regardless of the near parity of women representation among Argentinean and Brazilian herpetologists, women were marginalized within their co-author networks, due to the generalized preference of male authors to collaborate with other male authors (i.e., male homophily). Our findings indicate a pattern of male homophily, evidenced by the maximum author list length for exclusively male-authored papers being 13, compared to 5 for exclusively female-authored papers. This trend is further reflected in the frequency ranking of paper types, where the most common format is a paper authored by two men. In contrast, the equivalent female-only paper, authored by two women, ranks fourth and represents half the number of the two male-author papers. Additionally, papers authored by a single male occupy the sixth position in the ranking, whereas those authored by a single female are placed in the twelfth position. This aligns with the results of Fox et al. [23], who found that women are significantly underrepresented as sole authors compared to their representation in multi-authored papers.

Beyond these general patterns of data and homophily, our work is notable for considering the gender and positions of all co-authors. Our methodological proposal for measuring gender inequities deepens the approach by examining the roles of team members and quantifying authors' contributions to scientific papers by their harmonic weights [38]. Using harmonic weights corrects for inflationary and equalizing biases that can arise when authorship credit is allocated either by issuing full publication credit repeatedly to all coauthors, or by dividing one credit equally among all co-authors [38].

In our work, we considered that the last author had the lowest contribution along the author's list, which may be controversial due to the last author not always playing the same role [48] (see discussion in S2 File). We have categorized the potential errors based on how we assess the last author's contribution: either assuming the last author contributed more than the preceding authors (Type A) or assuming they contributed the least (Type B). Type A error occurs if we underestimate the last author's contribution, considering it poor when, in fact, they might be: A.i: A senior author who has contributed at least more than the preceding author on the list [30]. Type B error occurs if we overestimate the last author's contribution, considering it significant when they might be: B.i: A gifted author; B.ii: A guest author; B.iii: The one who contributed the least; B.iv: Someone positioned last due to their surname's initial letter being later in the alphabet than the preceding authors'; B.v: An author randomly positioned last. Given these scenarios, the most error-avoidant decision is to consider the last authors as having the least contribution [48, 49]. By not assuming that the last author is a senior author, we risk the opposite error: undervaluing their actual contribution. In those cases in which the last author acts as a group leader, she/he may be contributing to many works in parallel and thus their time investment must be distributed. This also supports the idea that the WCI can effectively measure the time invested in papers, as it considers both the number of authors in a paper and their position in the author list. Some research teams determine the order of author positions in an alphabetical listing based on the initial of the last name [50]. This particular practice is not of concern to our study, as in the event of numerous papers adopting this approach, the calculated WCI value would tend to resemble the expected chance value. Therefore, if it has any effect at all, it would likely lead to an underestimation of our result of women's contribution. Based on the aforementioned, the WCI provides a

valuable tool for quantifying individual contributions and understanding gender biases in academic publishing. We believe that our methodological proposal represents a reasonable new way of measuring gender bias able to capture broader information than previous methodological alternatives [10, 16, 50]. Moreover, the index could be customized to suit the specific question of interest or different assumptions of author inclusion and allocation based on additional information [38], for further arguments on how to value the author's contribution see the S2 File.

## Temporal trends within authorships

Our results showed that there was a notable temporal trend over the last 22 years, indicating a consistent rise in both the total number of authorships (from around 500 to 1000 per year) and women's authorships percentage in *Ecology* papers (from 25% to 35% of overall authorships). This aligns with the current trends observed in another ecological journal [29]. In addition, we found that mixed-gender papers increased during the study period, while papers written exclusively by men or women exhibited a declining trend over time. These patterns are in tune with the emergence and strengthening of modern scientific patterns, such as big science and technoscience [51], which are characterized by the exponential growth of multi-authored publications and larger team sizes [26, 52–54].

## Hierarchical organization of gender bias

We recognized three levels of gender bias that added evidence to the scaling pattern of gender bias in the global workforce [55]. The first level of bias we registered was the overall low proportion of women (i.e 30% of authors) publishing in *Ecology* during the period we analyzed. This magnitude matches the general trend of women participation already reported in other biological fields in several academic postgraduate communities from high-income countries [11, 13, 14, 50, 56, 57]. While it may be difficult to accurately estimate the global number of female ecologists in the academic realm, women represent 53% of bachelor's graduates, 43% of Ph.D. graduates, and only 28% of researchers in the field of ecology worldwide [58]. The second level of bias we found was that the female authorship percentage (accounting for the number of authoring events that disregard the author's identity) was lower than the female authors' percentages. We found a third level of bias when we analyzed the proportion of women among the authors who published most frequently in *Ecology*. The dearth of women becomes more pronounced in progressively more restrictive subsets of authors, who published more times in *Ecology* (i.e., top 100-50-25 authors). This represents a third bias level, with only 15% of women among the top 100 publishing authors between 1999 and 2021. Given that other authors reported up to 4% of women among the 100 top-publishing authors in *Ecology* in the 1945–2019 period [9], our results suggest an increase in women's representation among top publishing authors in the last 20 years. It is still unclear whether the frequency of women publishing in Ecology matches the rate of women making it to the list of top 100 authors.

The gender bias of the authors and authorship proportion and their decrease in top-publishing authors in *Ecology* shown in our work seems to be compatible with the hardening of academic demands: the higher the demands, the higher the gender bias [59]. A recent study by Andersson et al. [2] found that what we call requirement bias (i.e., the often implicit, higher demand in women's performance), increases as productivity increases. Also, it has been suggested that as researchers progress in their academic careers, requirement bias (unfavorable towards women) intensifies [55]. These increasingly greater difficulties in career advancement promote two main effects: 1- the impediment of women's advancement at the same rate of recruitment (for example when comparing Ph.D. and Senior researchers) [21] and commonly

referred to by the metaphor "glass ceiling" [55], and 2- the higher propensity of women to leave academia after their Ph.D. compared with their male colleagues [3] and commonly referred to by the metaphor "Leaky pipeline" [60, 61]. Both effects would point to the hierarchical organization of gender bias found in our data. In addition to these complex phenomena, we hypothesize another phenomenon to take into account: over-compensation.

### From over-contribution patterns to the over-compensation hypothesis

Our results from the WCI (Women's Contribution Index) showed a strong pattern of women's over-contribution in scientific publishing. This does not necessarily involve glorifying it as accomplishments of women who published in *Ecology*, despite their significant efforts. If doing, so we would be making a frequent statistical mistake known as *survivorship bias* [62, 63]. Survivorship bias, a form of selection bias, is a logical fallacy that involves focusing on the people or things that passed a selection process while overlooking those that did not, typically because they lack visibility. Survivorship bias leads to a more optimistic interpretation than the data offers and can lead to false conclusions in many different ways.

The over-contribution of women we observed could lead to a different interpretation of acknowledging it as an accomplishment. Instead of solely considering those "survivor" women who managed to publish in prestigious journals like *Ecology*, we can broaden the analysis to acknowledge that some women did not overcome the gender barriers. In this context, we could interpret the over-contribution pattern as arising from an attitudinal and psychological mechanism known as "*over-compensation*". Over-compensation was a term first proposed by psychologist Alfred Adler [64], that in the context of our study was coined merely conceptually. It involves the conscious or subconscious mechanism of concealing real or imaginary weaknesses, frustrations, inadequacies, or incompetence in one area of life by achieving excellence in another area. Thus, it is tightly associated with self-perception [64]. The higher proportion of women's contributions compared to what would be expected by chance suggests an adaptive strategy employed by female authors seeking to publish in *Ecology* papers.

Previously it has been suggested that women may over-compensate due to gender bias in the workplace [2, 20]. To overcome stereotypes and prove their competence, women may feel pressure to work harder, be more competent, and demonstrate their skills more strongly than men do. This phenomenon is colloquially referred to as the "prove-it-again", where women are evaluated more harshly and held to higher standards than men (i.e., *requirement bias*) [59]. In conjunction with the generally lower self-perception of women's abilities [17], the lower peer valuation of women can also trigger the phenomenon of over-compensation. In their seminal study, Moss-Racusin et al. [19] demonstrate that female candidates with equivalent academic qualifications for a technician position were perceived as less competent and less suitable for hire compared to male candidates. This suggests that a female scientist must surpass a male counterpart in performance to be considered comparable [20]. Moreover, Ross et al. [8] have recently studied the necessary level of work required for members of a research team to become an author, highlighting that it is more difficult for women than for men to be invited as co-authors. Thus, women must compensate for this bias with significantly more effort for their scientific contributions to be recognized. Thus, the female authors of *Ecology* may have had to exert more effort and invest more time in research than the average to become part of the research team behind a paper, as our WCI shows. The results of our work are largely compatible with the overcompensation hypothesis we are proposing. Future studies designed for this purpose will likely shed light on this potential phenomenon.

The women´s over-contribution pattern we found is prone to be a consequence of the higher dropout rates of women. Women in STEM fields have higher dropout rates than their

male counterparts [14, 65, 66]. Unfortunately, information regarding other biographic features of *Ecology* authors, such as their career stages, or the length of their academic trajectories was not available in our analyses. Thus, the bias we found towards women occupying the first author could be explained, at least partially, as a side effect of the presumably shorter careers of women authors of *Ecology*, which are dropped out by scientific pressures after publishing their first publication [14]. The rationale behind this involves that, early career researchers often occupy first author positions while seniors tend to occupy the last position (this is not always clear as discussed above).

However, there is a more complex scenario to consider, in which the dropout serves as both the cause and consequence of the observed differential contribution. One aspect of this differential gender contribution can be explained by the lower representation of women in advanced career stages compared to early stages, resulting from the dropout effect [14]. Related to the development of the career stage Manlove & Belou [67] described the proportion of lead authors, editorial board members, and editors of published manuscripts and found that lead authors with female names represent a large proportion of lead authors compared with editor positions. Additionally, the differential women's dropout effect itself could be attributed to over-compensation. This notion is based on the understanding that over-compensating requires a significant effort that becomes challenging to sustain over time, particularly for individuals facing not only inequalities related to career requirements (i.e. *requirement bias*) but also gender biases in other aspects (such as care tasks (i.e. *obstacle bias*) [21].

Future studies that consider both patterns of positions in the author list together with career trajectories could provide insights into the extent to which over-compensation influences gender-biased decisions to leave science (i.e., dropout).

## Final considerations

Our study's findings regarding the underrepresentation of female authors and their disproportionate high contribution to the Ecology Journal challenge the notion of science as an objective entity unaffected by biographic features, like gender. The idealized concept of science being objective often overlooks the historical context where science's practice and transmission were predominantly the domain of men, resulting in an inherently biased system [68–71]. Science's traditional foundations were shaped by and for white men, built upon ideals of objectivity, neutrality, and universality [69]. In this context, a theoretically meritocratic system may appear to align with the principles of this male-dominated scientific model. However, our findings support that the assumptions behind meritocracy do not hold and that survivor bias is responsible for masking the differential barriers to which people belonging to marginalized groups in academia are subjected.

Our results suggest that requirement bias may compel women in science to overcompensate, influencing their decisions to either persevere or exit the field. This possibility warrants further investigation through targeted experimental designs. Our research challenges the notion of meritocracy, allowing for a more comprehensive examination of the system compared to conventional approaches that overlook the experiences of marginalized women.

## Supporting information

**S1 File. Codes for women over-contribution analyses.** Printed version of the R markdown code for the data analysis and complete results in html format.
(HTML)

**S2 File. Methodological considerations for quantifying author contributions.** In this appendix, we present the rationale behind our methodological decision to consider the last author as the one who contributes the least. We believe this decision minimizes the likelihood of error in quantifying the last author contribution. Additionally, we propose potential methodological alternatives that the Women Contribution Index allows for when deemed appropriate by the researcher.
(PDF)

**S3 File. Alternative language abstract (Spanish).**
(PDF)

## Acknowledgments

We thank Jimena Grosso et al. [16] for providing us with the template for data acquisition; members of CienciaFem and other volunteers who helped in data collection; Jéssica Frattani helped us with the code; Instituto de Biodiversidad Neotropical (IBN) provided us a co-working space for the required meetings. We acknowledge the Postgraduate Secretary of the Faculty of Natural Sciences & Instituto Miguel Lillo, Universidad Nacional de Tucumán, for providing us with the physical space and institutional endorsement for the postgraduate course from which a large part of the results of this work derives. We also thank Claudia Noemi González Brambila, Joanna M Setchell, and one anonymous reviewer for helpful comments on the manuscript. We would like to express our gratitude to the Argentine public education and scientific systems and their policies aimed at strengthening science and gender equality in science. Additionally, we wish to voice our concern regarding the recent wave of governmental decisions negatively impacting Argentina's scientific system and infrastructure.

## Author Contributions

**Conceptualization:** Gabriela Fontanarrosa, Lucía Zarbá, Valeria Aschero, Daniel Andrés Dos Santos, María Gabriela Nuñez Montellano, Maia C. Plaza Behr, Natalia Schroeder, Silvia Beatriz Lomáscolo, Marisa Alvarez, Celina Reynaga, María Piquer-Rodríguez.

**Data curation:** Gabriela Fontanarrosa, Lucía Zarbá, Valeria Aschero, María Gabriela Nuñez Montellano, Maia C. Plaza Behr, Natalia Schroeder, Silvia Beatriz Lomáscolo, María Elisa Fanjul, A. Carolina Monmany Garzia, Marisa Alvarez, Agustina Novillo, María José Lorenzo Pisarello, Romina Elisa D'Almeida, Mariana Valoy, Andrés Felipe Ramírez-Mejía, Daniela Rodríguez, María Leonor Sandoval Salinas, Verónica Chillo.

**Formal analysis:** Gabriela Fontanarrosa, Lucía Zarbá, Valeria Aschero, Daniel Andrés Dos Santos, María Gabriela Nuñez Montellano, Maia C. Plaza Behr, Silvia Beatriz Lomáscolo, Agustina Novillo, Andrés Felipe Ramírez-Mejía, Celina Reynaga, María Piquer-Rodríguez.

**Funding acquisition:** Gabriela Fontanarrosa, María Gabriela Nuñez Montellano, María Piquer-Rodríguez.

**Investigation:** Gabriela Fontanarrosa, Lucía Zarbá, Valeria Aschero, Daniel Andrés Dos Santos, María Gabriela Nuñez Montellano, Maia C. Plaza Behr, Natalia Schroeder, María Elisa Fanjul, Marisa Alvarez, Agustina Novillo, María José Lorenzo Pisarello, Andrés Felipe Ramírez-Mejía, Verónica Chillo, María Piquer-Rodríguez.

**Methodology:** Gabriela Fontanarrosa, Lucía Zarbá, Valeria Aschero, Daniel Andrés Dos Santos, Andrés Felipe Ramírez-Mejía, María Piquer-Rodríguez.

**Project administration:** Gabriela Fontanarrosa, María Gabriela Nuñez Montellano, Maia C. Plaza Behr, Silvia Beatriz Lomáscolo, María Elisa Fanjul, María Piquer-Rodríguez.

**Supervision:** Daniel Andrés Dos Santos, María Piquer-Rodríguez.

**Visualization:** Gabriela Fontanarrosa.

**Writing – original draft:** Gabriela Fontanarrosa, Lucía Zarbá, Valeria Aschero, Natalia Schroeder, María Piquer-Rodríguez.

**Writing – review & editing:** Gabriela Fontanarrosa, Lucía Zarbá, Valeria Aschero, Daniel Andrés Dos Santos, María Gabriela Nuñez Montellano, Maia C. Plaza Behr, Natalia Schroeder, Silvia Beatriz Lomáscolo, María Elisa Fanjul, A. Carolina Monmany Garzia, Marisa Alvarez, Agustina Novillo, María José Lorenzo Pisarello, Romina Elisa D'Almeida, Mariana Valoy, Andrés Felipe Ramírez-Mejía, Daniela Rodríguez, Celina Reynaga, María Leonor Sandoval Salinas, Verónica Chillo, María Piquer-Rodríguez.

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
