## [Decision Letter · Decision Letter 0]

4 Apr 2024

PONE-D-24-00453Over twenty years of publications in Ecology: Over-contribution of Women reveals a new dimension of gender biasPLOS ONE

Dear Dr. Fontanarrosa,

Thank you for submitting your manuscript to PLOS ONE. After careful consideration, we feel that it has merit but does not fully meet PLOS ONE’s publication criteria as it currently stands. Therefore, we invite you to submit a revised version of the manuscript that addresses the points raised during the review process.

We look forward to receiving your revised manuscript.

Kind regards,

Claudia Noemi González Brambila, Ph.D.

Academic Editor

PLOS ONE

Journal Requirements:

Reviewers' comments:

Reviewer's Responses to Questions

**Comments to the Author**

1. Is the manuscript technically sound, and do the data support the conclusions?

Reviewer #1: Yes

Reviewer #2: Yes

2. Has the statistical analysis been performed appropriately and rigorously? 

Reviewer #1: Yes

Reviewer #2: Yes

3. Have the authors made all data underlying the findings in their manuscript fully available?

Reviewer #1: Yes

Reviewer #2: No

4. Is the manuscript presented in an intelligible fashion and written in standard English?

Reviewer #1: Yes

Reviewer #2: Yes

5. Review Comments to the Author

**Reviewer #1:** This manuscript investigates an important question - that of gender bias in publications in a STEMM subject - and introduces a new index to test for gender bias in the contribution to publications. The measure can easily to be extended to examine other biases, and is extremely useful. The dataset is impressive and the results are important and shocking, if unsurprising. The authors argue convincingly that they detect over-contribution by women, and link this to the inherently inequitable structure of the system we work in.

My only question, as I read the manuscript, is addressed in the discussion: the implications of the last author position being more regarded as of higher value than most other positions, at least in some cases (https://doi.org/10.1002/ece3.3435). The authors could present data on last author gender, perhaps compared with first author gender, but given that they include their data and code, others can explore these patterns, so this isn't necessary here.

I appreciate the clear statement of the limitations of the methods on p7.

Overall the manuscript is very easy to read. I recommend:

1. Ensuring that the figures are legible in greyscale

2. Checking that the figure captions match the figures, and that the text matches the figures. I could not match the text in LL290-3 to Fig 3, for example. The measured WCI in the text is 1307, but this doesn't match the violet line in Fig 3A. Can you direct the reader to a figure that shows that women's contributions stabilise at 0.3 to 0.35 (L282)?

3. The results of the KS test are missing (L316).

I also have some very minor comments on the clarity of phrasing.

L47: 'academia' (no initial capital)

L52: 'ponder' doesn't seem to be the right word here. Maybe 'measure', 'estimate' or 'quantify'?

L54, L57: 'with' works better than 'vs.'

L58: should Men's contribution index be in italics, to match L51?

L66: I think over-contribution has been explored, but not quantified.

L76: 'this' not 'it'

L87: it's easier to stick with one term than to tell the reader that two terms mean the same thing

L111: no need to refer to the discussion here.

L116: 'published in the journal *Ecology*' will do here**

L117: a Science, Technology, Engineering, and Mathematics (STEM)

L188: 'reported' not 'remarked'

L134: 'within the discipline' or 'within the field' but not both

L145: 'excluded' rather than 'dismissed'

L149: can you use a different word for 'non-occidental'? Later you use 'Global South', which works better.

L166: should 'Publication instances' also be in bold, like other variables?

L189: 'Figure 1 shows an example' is all you need

L205: I think A should be B - please check

L219: cut 'Noticeably'

L231: do you need the full explanation of qq plots? I think you can remove LL231-38.

L245: 'in' not 'under'

L266: no need for 'only'

L296: I was taught that we can never be 100% confident

L309: cut 'Furthermore'

L313: replace 'In the case where' with 'If'

L322: cut 'sex-segregated' because it's not quite accurate

L328: no need for 'and we discuss ...' because this is the discusssion

L344: 'pondered' is not the right word here

L381: replace 'during the analysed time-lapse' with 'during the period we analysed'

L387: 'found' not 'registered'

L388: is 'decayed' correct?

L389: I didn't understand this line.

L391: 'The dearth of women'

L392: no need for the italics

L397-8: can you clarify?

L412: there's a problem in the pdf I received here, with an orphan phrase

L431: maybe 'psychologist Alfred Adler'?

L437: cut 'sometimes' because you have 'may'

L444: 'classical' isn't quite right here

L454: cut 'side'

*L458: 'the bias we found' not 'the registered bias'*

***Reviewer #2:** Summary of the research*

The research uses descriptive statistics, and a novel index (WCI) to describe gender imbalances in author contribution to the journal Ecology. Following the finding that women are responsible for a higher WCI index than expected by chance, the authors reason that overcompensation (a consequence of gender discrimination), is responsible for the finding over overcontribution. This paper is well-written and the ‘from overcontribution to overcompensation’ theory is well argued. However, there are concerns over the presentation of speculations as conclusive evidence in the ‘final considerations’ section of the paper. A few terms, such as ‘contribution’ and ‘productivity’ could be better defined. The research would benefit from statistical tests (for example, on the change in author compositions over time), and more generally by emphasizing findings from the descriptive statistics which are persuasive for a need to explore gender bias in this field, rather than an overwhelming focus on the WCI which has numerous limitations as an index of contribution.

Major comments

The design of the index to describe gender contributions [L169-L207] is novel and provides an interesting contribution to highlight gender imbalances in the wider field. However, the decision to assume that the last author has the least contribution is controversial. The authors justify this contentious decision [L348], elaborate on the limitations that stem from it, and subsequently acknowledge that the last author tends to represent the senior researcher [L462]. Although meanings of authorship positions vary between disciplines, there is strong evidence that in the field of Ecology, most people view the last author as a senior author. For example: Fox, Ritchey and Paine (2018); Duffy (2017); Weltziin et al. (2006). Furthermore, there is a tendency to perceive the ‘corresponding’ author as having a higher contribution to the research. In its current form, the Women’s Contribution Index does not adequately represent the contribution of the final and/or corresponding author. Attributing the final author a proportionally larger contribution than the preceding authors or taking account of the identity of the corresponding authors, are both possible adjustments that would enhance the validity of the index.

[L290 – L293] The stats comparing observed and expected WCI values in this section of the results are confusing to read. The observed WCI value is stated as a ‘total sum value’, and it is presented adjacent to a ‘mean’ expected value, and then a ‘maximum’ expected value. The conclusions of the paper rest on the finding that the simulated WCI value “did not exceed the observed value of WCI” [L294]. This is currently of great concern as the stated observed value of WCI is 1,307 [L291]. The stated expected WCI value is 1,361, with a maximum of 1,388 [L293]. As currently stated, the observed value is less than the expected value (1,307 < 1,361). Figure 3A suggests an observed value of c1450 from reading the value of the purple vertical line. Is the observed value of the WCI quoted incorrectly in this paragraph?

[L480-L495] The paper is mostly consistent in presenting ‘overcompensation’ as a theory to explain the finding of women’s overcontribution. However in the final considerations section, overcompensation is presented as a cause of overcontribution as though there is conclusive evidence for this theory. This does not follow coherently from the body of the paper. Without presenting ‘overcompensation causing overcontribution’ as a theory that remains to be tested, this paper runs the risk of appearing to satisfy a confirmation bias. The abstract sets out an intention to ‘detect potential gender biases in the authorship of papers and examine the extent of women’s contribution to… scientific production in Ecology’ [L48-L50]. The alternative hypothesis [L217] that the observed WCI value differs from the expected WCI value can be met by either woman’s contribution (i) being lower than expected, or (ii) higher than expected. The descriptive statistics very clearly show women are underrepresented: there are fewer female authors (L255: 31.94%), women account for fewer authorship events (L256: 27.78%), and women have lower average publication instances (L257: 1.41 versus 1.72 for men). If the WCI value was observed to be lower than expected, presumably this would have been interpreted as an example of underrepresentation stemming from discrimination. Rejecting the null hypothesis of no difference in observed versus expected WCI values and arriving at the same conclusion (that women are experiencing discrimination in this field), is potentially concerning. The argument of survivorship bias, and the theory of overcompensation causing overcontribution is convincing, but it is strongly recommended that this theory is presented consistently as a theory, not as a conclusion. The authors make no other suggestions to explain women’s overcontribution, which seems amiss.

In the current state of the paper the quoted observed WCI value is less than the expected value, and the WCI suffers from a potential de-valuing of the last (senior) authorship position. The descriptive statistics alone are persuasive that there is a need to address gender bias in this field. For example, the section on hierarchical organisation of gender bias in the discussion [L378-L399] is very well-written and is very persuasive. The paper could benefit from emphasizing known entities (descriptive stats) rather than speculating on more exploratory stats such as the WCI value.

As the authors acknowledge in the discussion [L457], this research would benefit from including other biographic features. The inclusion of career stages and length of academic trajectories could greatly influence the number of authorship events per author, and position in an author list. I look forward to future efforts to explore this question with this in mind.

Minor comments

The paper repeatedly refers to ‘productivity’ but has not defined productivity. For example, [L128]: ‘a gender inequity of total productivity that exceeds 35%’, [L332] ‘gender disparities in … productivity’, and [L402] ‘requirement bias… increases as productivity increase’. Productivity can mean many things and is not sufficiently defined.

[L93] What is the meaning of ‘stimulation’, in this context?

[L94] “Support” needs the word ‘less’ added in-front, to read ‘less support’.

[L99] “Also true for other genders”. This feels tokenistic. Suggest that either the paper should address the biases experienced by other genders, or this should be removed altogether as it is not the focus of the paper to look at non-binary contributions to, or the biases experienced by non-binary people, in academia.

[L123] Methods. It would be beneficial to know the authors reasons for selecting the field of the ecology and specifically the journal of ecology, as case studies. Are the authors from this field? Do they expect this field to be representative of the wider STEM discipline?

[L130]. “Moreover” can be removed, as presumably literature in physics is also dominated by male scientists mainly from North America and Europe.

[L140] The gender checker process seems robust.

[L146] How many papers were dismissed using this process?

[L158-L167] Gender data overview. The descriptive statistics are well set out and easy to comprehend.

[L209]. Unbiased simulation. This section needs clarity: Did the authors start with fixed ratios of women and men in each authorship list, per paper, and randomise within set authorship lists; or did they compile all female and male names together and then run simulations across all lists combined?

[L224] Define reason for splitting the data set in two subsets: presumably to avoid dependency in the data?

[L252] Typo ‘o’ should be ‘or’.

[L258] Interesting to read that the most common paper type was authored by two males. This could be set in more context of the other common paper types (number of authors and composition).

[L281] Are there any statistical results to show an increase in women’s authorship over time? The trend stabilising in a ratio of women/ total authors around 0.3 and 0.35 is confusing. Do they mean a ratio of women:total authors of 0.3? Why is it 0.3 and 0.35?

[L356] The argument that the last author should not be attributed higher contribution because even if he/she is a senior member they will have more distributed time investment is confusing. The assumptions in this argument for attributing the final author the smallest contribution are as follows: (i) where last author is senior, (ii) seniors will be contributing to many works, (iii) therefore their time is distributed, (iv) therefore they have given the least amount of time to this work. At no point have the authors defined ‘contribution’ as synonymous with ‘time investment’. Contribution can come in many forms; research conception, design, data interpretation, drafting, revisions and guidance to all of these. The time taken to contribute to each of these areas may vary, so the argument that a senior author contributed less because their time is assumed to be more taxed is not strong. Contribution’ is not well described at any point, other than being reflected by author positions in the WCI.

[L367-L371] Discussion of temporal trends would benefit from statistical tests showing the significance of changes in total authorships and women’s authorships over time.

References

Fox, C. W., Ritchey, J. P., & Paine, C. T. (2018). Patterns of authorship in ecology and evolution: First, last, and corresponding authorship vary with gender and geography. Ecology and evolution, 8(23), 11492-11507

Weltzin, J. F., Belote, R. T., Williams, L. T., Keller, J. K., & Engel, E. (2006). Authorship in ecology: Attribution, accountability, and responsibility. Frontiers in Ecology and the Environment, 4(8), 435–441. https://doi.org/10.1890/1540-9295(2006)4[435:AIEAAA]2.0.CO;2

*Duffy, M. A. (2017). Last and corresponding authorship practices in ecology. Ecology and Evolution, 7(21), 8876–8887. https://doi.org/10.1002/ece3.3435*

*6. PLOS authors have the option to publish the peer review history of their article (what does this mean?). If published, this will include your full peer review and any attached files.*

**

Reviewer #1: Yes: Joanna M Setchell

Reviewer #2: No

**

*While revising your submission, please upload your figure files to the Preflight Analysis and Conversion Engine (PACE) digital diagnostic tool, https://pacev2.apexcovantage.com/. PACE helps ensure that figures meet PLOS requirements. To use PACE, you must first register as a user. Registration is free. Then, login and navigate to the UPLOAD tab, where you will find detailed instructions on how to use the tool. If you encounter any issues or have any questions when using PACE, please email PLOS at figures@plos.org. Please note that Supporting Information files do not need this step.*

---

## [Author Response · Author response to Decision Letter 0]

3 Jun 2024

Rebuttal letter for PONE-D-24-00453

Over twenty years of publications in Ecology: Over-contribution of Women reveals a new dimension of gender bias

Dear Editor: 

Each reviewer's comment is addressed individually, providing detailed responses and explanations for the changes made.

Reviewer #1: This manuscript investigates an important question - that of gender bias in publications in a STEMM subject - and introduces a new index to test for gender bias in the contribution to publications. The measure can easily to be extended to examine other biases, and is extremely useful. The dataset is impressive and the results are important and shocking, if unsurprising. The authors argue convincingly that they detect over-contribution by women, and link this to the inherently inequitable structure of the system we work in.

My only question, as I read the manuscript, is addressed in the discussion: the implications of the last author position being more regarded as of higher value than most other positions, at least in some cases (https://doi.org/10.1002/ece3.3435). The authors could present data on last author gender, perhaps compared with first author gender, but given that they include their data and code, others can explore these patterns, so this isn't necessary here.

I appreciate the clear statement of the limitations of the methods on p7.

Our response: Thank you very much.

Overall the manuscript is very easy to read. I recommend:

1. Ensuring that the figures are legible in greyscale

Our response: Done

2. Checking that the figure captions match the figures, and that the text matches the figures. I could not match the text in LL290-3 to Fig 3, for example. The measured WCI in the text is 1307, but this doesn't match the violet line in Fig 3A.

Our response

We are deeply sorry for this big mistake and thankful to the reviewer for pointing this out. The error proceeds from a non-well-checked mix of versions of our manuscript. In an earlier version with a smaller dataset, the total sum WCI was 1307, whereas in the current version with the full dataset, the total sum WCI is 1456. We corrected the value in the manuscript and double-checked there were no other mistakes like this one in the results and figures. Furthermore, for more clarity, we included in the manuscript a table with key summary values (Table 1).

Now the text is written: “We measured the WCI for each paper in our dataset and found that the total sum value for all papers was 1,456. Upon conducting the randomized simulations, we obtained a simulated WCI distribution with a mean value of 1,361 and a standard deviation of 7.75. Notably, even the maximum value in the WCI distribution after running the 10,000 randomized simulations (1,388) did not exceed the observed value of WCI (vertical violet line in Fig. 3A).”

 Can you direct the reader to a figure that shows that women's contributions stabilize at 0.3 to 0.35 (L282)?

Our response: It referred to figure 2 D (bottom) We rephrased the sentence to: “Starting around 2012, the ratio of women to total authors appears to stabilize around 0.3 to 0.35” 

3. The results of the KS test are missing (L316).

Our response: We now included the results in the text: “Furthermore, using a Kolmogorov-Smirnov test, we found that the compared distributions were significantly different (D = 0.13971, p-value < 2.2e-16).” 

Dear Joanna Setchell, 

We are very grateful to you for the thoughtful comments you made about our work. We took almost all your suggestions and appreciated your careful reading of our paper, feeling that they were raising the quality of the manuscript. 

The Last Author Issue (this response is replicated for both reviewers as highlighted the same point to us)

As you can imagine, assessing the contribution of the last order became a controversial issue, igniting an intriguing internal debate within our research team. After considering other methodological options we concluded that taking the last author as the one who contributes the least is the best option, although not a perfect one. We argue that this methodological decision is the one that deals with less problematic assumptions. 

We have categorized the potential errors based on how we assess the last author's contribution: either assuming the last author contributed more than the preceding authors (Type A) or assuming they contributed the least (Type B).

Type A Error:

This error occurs if we underestimate the last author's contribution, considering it poor when, in fact, they might be:

A.i: A senior author who has contributed at least more than the preceding author on the list.

Type B Error:

This error occurs if we overestimate the last author's contribution, considering it significant when they might be:

B.i: A gifted author (details provided below).

B.ii: A guest author (details provided below).

B.iii: The one who contributed the least.

B.iv: Someone positioned last due to their surname's initial letter being later in the alphabet than the preceding authors'.

B.v: An author randomly positioned last.

Given these scenarios, the most error-avoidant decision is to consider the last authors as having the least contribution. By not assuming that the last author is a senior author, we risk the opposite error: undervaluing their actual contribution. However, in cases where the last author is indeed a senior author, this issue is less severe since senior authors are typically very busy individuals whose time is divided among many projects. It is important to remember that the WCI aims to capture time investment accurately.

 (We have now reinforced this idea for greater clarity in Materials and Methods: “We aim to capture the authors' degree of contribution to a paper in terms of time investment”).

The following items explain our arguments regarding the inconvenience of assuming the last author consistently guides the research teams and contributes more than the previous one in the list. 

1) Temporal trends regarding the last position occupancy. Our data are dated from 1999 to almost the present. Historically, there was consensus that the contribution was decreasing in the list of authors. Only recently did the latest authors take center stage, being praised as seniors. Currently, both strategies coexist. Considering the last author as a senior is not parsimonious enough. 

Duffy et al. (2017) found in a survey that most ecologists view the last author as the “senior” author on a paper (i.e., the person who guides the research group in which most of the work was carried out). However, there was substantial variation in views on authorship, especially corresponding authorship. In 2016, the corresponding author was usually the first author (range across the four journals: 77%–90% of papers); less commonly, it was the last author (range across the four journals: 9%–18% of papers. 

In the manuscript, we state: “The last author may or not reflect an advisory role. Nevertheless, a co-authorship position may also imply other or even arbitrary decisions (...)”

2) More than one senior author may be co-authors in a paper. Under this situation, it is not easy (even impossible with our data) to inquire regarding the positions of multiple senior authors.

3) It is difficult to establish at what number of authors the last author becomes a significant contributor. We wonder at what number of authors does the last one become a considerable leader? 

4) The Gift Authorship and Guest Authorship problem: 

Laboratory group leaders or other senior academics are prone to be gifted authors. Authorship gifting happens when an individual is acknowledged in a study but doesn't meet the criteria for authorship. This practice is also referred to as honorary authorship. Essentially, it's a gesture; the individual doesn't qualify as an author per se. 

 The Guest Authorship. The laboratory group leader or another senior academic is prone to be gifted authors. Guest authorship occurs when influential individuals "loan" their name to a study to enhance its credibility. Nevertheless, these individuals were not directly involved in the research itself. One of the main drivers of guest authorship stems from the hierarchical organization of contemporary laboratories. For instance, principal investigators frequently require that their names be included or listed first on research conducted within their department or laboratory. They assert this demand based on their acquisition of research funds or their provision of top-level supervision. Both Gift Authorship and Guest Authorship are related to the Matthew effect. Both gift and guest authorship are commonly accepted by the actual authors partly because it's widely understood that their paper stands to benefit in terms of acceptance if they include the name of a renowned researcher. This situation adds extra noise to our attempts to understand roles in academia given that many laboratory heads and senior academics are male, they are overrepresented given the glass ceiling of the leaky pipeline. 

Alternatives adjustments to the WCI: Our approach follows the Harmonic Allocation of Authorship Credit proposed by Nils Hagen in 2008 (doi:10.1371/journal.pone.0004021) for measuring the contribution of multiple authors in a paper. What we contribute to building the WCI is tallying only the contributions made by female authors; in other words, other researchers might employ different methods to measure contributions in multi-authored papers and calculate the WCI. We invite future studies to propose enhancements in index construction. As possible correction can still use some version of the Harmonic Allocation of Authorship Credit (Hagen, 2008).

If we assume that the last author is the team leader of a particular paper: Which would be the best way to value their weight? Equal to the first author? Equal to the second author? Both possibilities may be ok. Another possibility could be to consider the weight of the last author as the average weight of all other authors, and then distribute the remaining weight among the remaining authors using harmonic decay.

Hagen (2008) discusses the possibility of including additional byline information about the equality of some co-authors' contributions, or implicit information about the approximate equality of contributions by the first and last authors. Such variations are easily accommodated by a harmonic counting scheme with little or no alteration of the credit allocated to the remaining coauthors (see Hagen Figure 5). Although the Harmonic Allocation of Authorship Credit can deal with, for example, first and last authors equally merited (proposed by Hagen itself), we still think our study case prevents us from doing so. For further clarifications see arguments in Hagen (2008). 

We agree that the corresponding author information is valuable. Frequently used to indicate in which situations the last author is the team leader; unfortunately, we did not register them in our data set. Future research may include them after a few changes in the Women Contribution Index. Nevertheless, due to Duffy (2017) finding that 84% of papers published in 2016 had the first author as the corresponding author, we think that our results are somehow capturing the corresponding author's trends.

-To clarify this last author issue we have reinforced the argumentative line in the manuscript section: Measuring gender inequities (Discussion).: “We have categorized the potential errors based on how we assess the last author's contribution: either assuming the last author contributed more than the preceding authors (Type A) or assuming they contributed the least (Type B). Type A error occurs if we underestimate the last author's contribution, considering it poor when, in fact, they might be: A.i: A senior author who has contributed at least more than the preceding author on the list (Duffy, 2017). Type B error occurs if we overestimate the last author's contribution, considering it significant when they might be: B.i: A gifted author; B.ii: A guest author; B.iii: The one who contributed the least; B.iv: Someone positioned last due to their surname's initial letter being later in the alphabet than the preceding authors'; B.v: An author randomly positioned last. Given these scenarios, the most error-avoidant decision is to consider the last authors as having the least contribution (Tarkang et al. 2017; Fernandes et al, 2020). By not assuming that the last author is a senior author, we risk the opposite error: undervaluing their actual contribution. In those cases in which the last author acts as a group leader, she/he may be contributing to many works in parallel and thus their time investment must be distributed (.....).Moreover, the index could be customized to suit the specific question of interest or different assumptions of author inclusion and allocation based on additional information (see [36]), for further arguments on how to value the author's contribution see the S2 Text.” 

We have included a Supporting Information text (S2 Text) that further discusses the last author's issue and where we propose some alternative ways of considering the Last Authors.

 An extra discussion regarding the last author:

Fox, Ritchey, and Paine (2018) measure the proportion of the last author based on gender, assuming the last author is a senior. They based this assumption on the paper of Duffy, 2017. They found that Women were less likely to be last (for them considered to be “senior”) authors (~23%) and sole authors (~24%), but more likely to be first authors (~38%), relative to their overall frequency of authorship (~31%). 

 Grosso et al., 2021 also found a similar pattern in the field of Herpetology. This recurrence of patterns seems to show a non-random process behind who occupies the last author position. If we assign the last authors with great value, probably the value of the WCI will be lower than what we found. Even so, given that the value of WCI that we found is larger than the highest random value in 10,000 simulations, probably the value of WCI observed if we considered the last car with the highest values would most likely still be higher than the value expected by chance. Nevertheless, it is important to take into account that, by doing so we also would be introducing too much noise to sour analysis. Gift authors and Guest authors will probably occupy the last authorship. Here we reinforce the idea that our index measures the time investment in the papers. 

I also have some very minor comments on the clarity of phrasing.

Our response: All the following minor comments were incorporated. We only added clarifications itemized for those comments where it is appropriate (see below).

L47: 'academia' (no initial capital)

L54, L57: 'with' works better than 'vs.'

L58: should Men's contribution index be in italics, to match L51?

L66: I think over-contribution has been explored, but not quantified.

L76: 'this' not 'it'

L111: no need to refer to the discussion here.

L116: 'published in the journal Ecology' will do here

L117: a Science, Technology, Engineering, and Mathematics (STEM)

L188: 'reported' not 'remarked'

L134: 'within the discipline' or 'within the field' but not both

L145: 'excluded' rather than 'dismissed'

L166: should 'Publication instances' also be in bold, like other variables?

L189: 'Figure 1 shows an example' is all you need

L219: cut 'Noticeably'

L231: do you need the full explanation of qq plots? I think you can remove LL231-38.

L245: 'in' not 'under'

L266: no need for 'only'

L309: cut 'Furthermore'

L313: replace 'In the case where' with 'If'

L322: cut 'sex-segregated' because it's not quite accurate

L328: no need for 'and we discuss ...' because this is the discussion 

L344: 'pondered' is not the right word here

L381: replace 'during the analysed time-lapse' with 'during the period we analysed'

L387: 'found' not 'registered'

L391: 'The dearth of women'

L392: no need for the italics

L431: maybe 'psychologist Alfred Adler'?

L437: cut 'sometimes' because you have 'may'

L444: 'classical' isn't quite right here /// Our response: It was changed by “In their seminal study, Moss-Racusin et al. [19] demonstrate that”

L454: cut 'side'

L458: 'the bias we found' not 'the regis

---

## [Decision Letter · Decision Letter 1]

12 Jul 2024

Over twenty years of publications in Ecology: Over-contribution of Women reveals a new dimension of gender bias

PONE-D-24-00453R1

Dear Dr. Piquer-Rodriguez,

We’re pleased to inform you that your manuscript has been judged scientifically suitable for publication and will be formally accepted for publication once it meets all outstanding technical requirements.

Kind regards,

Claudia Noemi González Brambila, Ph.D.

Academic Editor

PLOS ONE

Additional Editor Comments (optional):

Reviewers' comments:

Reviewer's Responses to Questions

**Comments to the Author**

1. If the authors have adequately addressed your comments raised in a previous round of review and you feel that this manuscript is now acceptable for publication, you may indicate that here to bypass the “Comments to the Author” section, enter your conflict of interest statement in the “Confidential to Editor” section, and submit your "Accept" recommendation.

Reviewer #2: All comments have been addressed

2. Is the manuscript technically sound, and do the data support the conclusions?

Reviewer #2: Yes

3. Has the statistical analysis been performed appropriately and rigorously? 

Reviewer #2: Yes

4. Have the authors made all data underlying the findings in their manuscript fully available?

Reviewer #2: Yes

5. Is the manuscript presented in an intelligible fashion and written in standard English?

Reviewer #2: Yes

6. Review Comments to the Author

Reviewer #2: The authors addressed all of the initial review comments very well. Their responses were thoughtful and detailed. They have provided an in depth justification for why they are not changing the weight of the final author - which addresses the key criticism from both reviewers. The final manuscript has addressed many areas that were previously unclear and has corrected a few errors reporting statistical values. The addition of Table 1 is useful, although the formatting is not the easiest visually to interpret.

7. PLOS authors have the option to publish the peer review history of their article (what does this mean?). If published, this will include your full peer review and any attached files.

Reviewer #2: No

---

## [Editor Report · Acceptance letter]

2 Aug 2024

PONE-D-24-00453R1 

PLOS ONE

Dear Dr. Piquer-Rodriguez, 

I'm pleased to inform you that your manuscript has been deemed suitable for publication in PLOS ONE. Congratulations! Your manuscript is now being handed over to our production team.

Kind regards, 

on behalf of

Dr. Claudia Noemi González Brambila 

Academic Editor

PLOS ONE